# Association between Workplace Bullying, Job Stress, and Professional Quality of Life in Nurses: A Systematic Review and Meta-Analysis

**DOI:** 10.3390/healthcare12060623

**Published:** 2024-03-09

**Authors:** Petros Galanis, Ioannis Moisoglou, Aglaia Katsiroumpa, Maria Mastrogianni

**Affiliations:** 1Clinical Epidemiology Laboratory, Faculty of Nursing, National and Kapodistrian University of Athens, 11527 Athens, Greece; aglaiakat@nurs.uoa.gr; 2Faculty of Nursing, University of Thessaly, 41500 Larissa, Greece; iomoysoglou@uth.gr; 3Faculty of Nursing, National and Kapodistrian University of Athens, 11527 Athens, Greece; mamastr@nurs.uoa.gr

**Keywords:** workplace bullying, job stress, professional quality of life, nurses, meta-analysis

## Abstract

Workplace bullying affects workers’ lives, causing several mental and physical health problems and job-related issues. Therefore, a summary of the evidence on the consequences of workplace bullying on workers’ lives is essential to improve working conditions. The literature lacks systematic reviews and meta-analyses on the association between workplace bullying and job stress and the professional quality of life of nurses. Thus, we aimed to quantitatively summarize the data on the association between workplace bullying, job stress, and professional quality of life. We performed our study in accordance with the Preferred Reporting Items for Systematic Reviews and Meta-Analysis guidelines. The review protocol was registered with PROSPERO (CRD42024495948). We searched PubMed, Medline, Scopus, Cinahl, and Web of Science up to 4 January 2024. We calculated pooled correlation coefficients and 95% confidence intervals [CI]. We identified nine studies with a total of 3730 nurses. We found a moderate positive correlation between workplace bullying and job stress (pooled correlation coefficient = 0.34, 95% CI = 0.29 to 0.39). Moreover, a small negative correlation between workplace bullying and compassion satisfaction (pooled correlation coefficient = −0.28, 95% CI = −0.41 to −0.15) was identified. Additionally, our findings suggested a moderate positive correlation between workplace bullying and job burnout (pooled correlation coefficient = 0.43, 95% CI = 0.32 to 0.53) and secondary traumatic stress (pooled correlation coefficient = 0.36, 95% CI = 0.11 to 0.57). Our findings can help nursing managers and policy-makers to draw attention to workplace bullying by implementing effective interventions, so as to reduce the bullying of nurses.

## 1. Introduction

Even if the definition of workplace bullying differs significantly between studies, workplace bullying refers to a situation where a worker is, over some time period, exposed to persistent negative mistreatment, in both the private and public sectors, consisting of frequent criticism and person-related physical, verbal, or psychological violence [1,2]. The literature suggests that there are consequences of workplace bullying, i.e., mental health problems (i.e., burnout, anxiety, depression, posttraumatic stress disorder) [3], several physical problems (i.e., muscle pain, fatigue, headache, angina, hypertension) [4], and job-related issues (e.g., decline in task performance, lower job satisfaction, poor productivity) [5].

The occurrence of workplace bullying against nurses is a constant and extensive worldwide phenomenon, but its exact level is difficult to determine [6,7]. Compared with other healthcare professionals, nurses confront a higher degree of workplace bullying [8]. Recent systematic reviews report an average rate of 26.3% [3], and that the rate may even reach as high as 94.0% [9]. About 22.0–44.0% of nursing professionals have faced some kind of workplace bullying during their career [10,11], with 42.1–44.0% having experienced it in the last 12 months [12]. In some studies, workplace bullying was associated with individual status, like gender, academic level, and family situation [9,13,14], as well as organizational structure and management style [15]. As a special type of workplace violence, it can cause several physical and psychological issues, including fear, loss of sleep, anxiety, uncertainty, nightmares, anger, apathy, embarrassment, burnout, depression, fragility, vulnerability, lack of confidence, and suicidal ideation [3,9,13,16,17]. In many cases, when the negative effects are too hard for nurses to handle, workplace bullying is one of the main reasons for their intention to quit [18], as it is accompanied by two chronic phenomena, job stress and low professional quality of life.

Stress can be defined as any physical or psychological tension generated by either external or internal factors, while job stress, also known as occupational stress, occurs when a nursing professional cannot cope with job demands, and thus, there is a negative impact on them and/or the workplace [19]. The literature suggests several risk factors for job stress in nurses, such as job burnout, insufficient job control, lack of professional nursing competence, job dissatisfaction, low personal accomplishment, verbal abuse, heavy workload, personal conflict, work in high-risk units (e.g., psychiatric intensive care units), and physical violence [20,21,22,23]. According to the findings of a systematic review, it is estimated that the costs of stress at the national level in high-income countries (Australia, Denmark, France, Sweden, and the United States) range from USD 221 million to USD 187 billion and account for medical (direct) and production-related (indirect) costs. According to the same systematic review, 70–90% of the total costs are related to loss of productivity, i.e., sickness absence, premature death, and premature workforce exit [24]. The similar cost for China exceeds USD 900 million (Hong Kong USD 7.09 billion) [25].

Work-related stress may take three different forms: acute, episodic, and chronic work-related stress [26]. Acute stress, the most common, happens when somebody cannot easily handle situations that an average person can, episodic stress occurs when acute stress takes place frequently, and chronic stress occurs when the period of the phenomenon occurrence is long-lasting. Job stress is one of the most frequent consequences of workplace bullying with numerous negative effects, such as physical and mental disorders, decreased work-related satisfaction, decreased efficiency, and lower quality of health-related care, which in turn leads to lower professional quality of life [27].

Workplace bullying influences nurses’ professional quality of life (ProQOL) [28], which demonstrates the professional worker’s quality of life for those who work as a helper [29]. In order to measure one’s professional quality of life, three aspects must be taken into consideration: compassion satisfaction, burnout, and secondary trauma stress [29]. Compassion satisfaction is considered to be the emotional, psychological, and physical exhaustion caused by chronic job-related stress [30]. Burnout refers to a negative sentimental response to external stressors linked with working conditions [31]. Secondary traumatic stress is an emotional reaction combined with vulnerability to another’s stressful experience [32]. Moreover, burnout and secondary trauma stress may influence patients’ outcomes, as a low ProQOL may lead to a reduced interest in patients, lower job productivity, increased practice errors, and team deterioration [33,34,35].

The literature indicates that workplace bullying is associated with job stress and the professional quality of life of nurses, but the degree of association is unclear. Several systematic reviews investigated the impact of workplace bullying caused by colleagues on turnover intention, burnout, post-traumatic stress disorder, psychological distress, anxiety, depressive symptoms, and the general health of nurses [15,36,37,38,39]. However, no systematic reviews have been conducted thus far on the association between workplace bullying and job stress and the professional quality of life of nurses. Moreover, two systematic reviews [40,41], one narrative review [42], one scoping review [43], and one umbrella review [44] have already examined the impact of workplace bullying caused by patients and relatives on nurses. The literature suggests that workplace bullying caused by patients and relatives increases turnover intention, absenteeism, burnout, job dissatisfaction, post-traumatic stress disorder, stress, anxiety, depression, and fear among nurses. The most common types of workplace bullying caused by patients and relatives that nurses face are verbal abuse and physical abuse. Thus, in view of this situation, we aimed to quantitatively summarize the data on the association between workplace bullying and job stress and the professional quality of life of nurses. We focused our study on workplace bullying caused by colleagues rather than on workplace bullying caused by patients and/or relatives.

## 2. Materials and Methods

### 2.1. Data Sources and Strategy

We applied the Preferred Reporting Items for Systematic Reviews and Meta-Analysis (PRISMA) guidelines [45] to perform our review and meta-analysis (Appendix A). We searched PubMed, Medline, Scopus, Cinahl, and Web of Science for articles dated from their inception to 4 January 2024. Our research questions were the following: (a) Is there any association between workplace bullying and the job stress of nurses? (b) Is there any association between workplace bullying and the professional quality of life of nurses? Furthermore, we expanded these research hypotheses by conducting a meta-analysis to quantify the association between our study variables. We searched in article title/abstract using the following strategy: ((“workplace bullying” OR bullying OR “workplace violence” OR violence OR “horizontal violence” OR “internal violence”) AND (nurs*)) AND (“job stress” OR stress OR “work stress” OR “occupational stress” OR “job anxiety” OR anxiety OR “job distress” OR distress OR “professional quality of life” OR “job quality of life” OR “work quality of life” OR “quality of life”). The review protocol was registered with PROSPERO (CRD42024495948).

### 2.2. Selection and Eligibility Criteria

Two independent reviewers performed the study selection, while a senior reviewer resolved the disagreements. First, we removed duplicates, and then we screened titles, abstracts, and full texts. We applied the following inclusion criteria: studies that examined the association between workplace bullying caused by colleagues, job stress, and professional quality of life; studies that included nurses working in clinical settings; quantitative studies; studies that were published in the English language; studies that used valid instruments to measure workplace bullying, job stress, and professional quality of life. We excluded qualitative studies, case reports, meeting or conference abstracts, protocols, editorials, letters to the Editor, reviews, and meta-analyses. Moreover, we excluded studies that included data for healthcare workers in total without separating them for nurses. Additionally, we excluded studies that examined workplace bullying caused by patients and/or relatives.

### 2.3. Data Extraction

Two independent authors performed the data extraction procedure, while a third author resolved discrepancies. We extracted the following data from the studies: authors, country, data collection time, females’ percentage, age, sample size, study design, sampling method, clinical settings, assessment tools for workplace bullying, job stress, professional quality of life, and response rate. Moreover, to perform the meta-analysis, we extracted measures of effect for the association between workplace bullying, job stress, and professional quality of life: correlation coefficients, regression coefficients beta from linear regression models, and odds ratios from logistic regression models.

### 2.4. Quality Assessment

We used the Joanna Briggs Institute critical appraisal tools [46] to examine the quality of our studies. This risk-of-bias tool includes eight different assessment domains, such as study settings, inclusion criteria, exposure and outcome assessment, elimination of confounders, etc. The quality of studies was classified as poor (score of ≤3 points), moderate (score of 4–6 points), or good (score of 7–8 points) according to an 8-point scale.

### 2.5. Statistical Analysis

In our review, six studies out of nine presented correlation coefficients between workplace bullying, job stress, and professional quality of life. Moreover, five studies presented unstandardized regression coefficients beta from linear regression models, while one study presented odds ratios from logistic regression models. Among the studies that performed a linear regression analysis, only one study provided 95% confidence intervals (CIs) for unstandardized coefficients beta, allowing us to calculate standard errors and standardized coefficients beta. Thus, we cannot perform a meta-analysis using data from linear and logistic regression models. However, sufficient data for correlation coefficients enabled us to perform a meta-analysis. In this case, we calculated the pooled correlation coefficients between workplace bullying, job stress, and professional quality of life and the 95% CIs. Cohen suggests the following cut-off points for correlation coefficients: 0.10 to 0.29 or −0.29 to −0.10 indicates a small effect; 0.30 to 0.49 or −0.49 to −0.30 indicates a moderate effect; 0.50 to 1.0 or −1.0 to −0.50 indicates a large effect [47].

We used the I^2^ statistics and the *p*-value for the Hedges Q statistic to examine heterogeneity in our meta-analysis [47]. In this case, I^2^ values >75% are indicative of high heterogeneity, while a *p*-value less than 0.1 for the Hedges Q statistic is indicative of statistically significant heterogeneity [48]. When the heterogeneity between results was high, we applied a random effect model to estimate the pooled correlation coefficient, while when heterogeneity was low, we applied a fixed effect model. Additionally, we performed a leave-one-out sensitivity analysis to estimate the impact of each study on the overall correlation coefficient. We examined sources of heterogeneity by applying a meta-regression analysis for continuous variables. In that case, we considered the following variables as sources of heterogeneity: data collection time, females’ percentage, age, sample size, and response rate. Due to limited variability among our studies, country, study design, sampling method, clinical settings, assessment tools, and quality of studies could not be considered as potential sources of heterogeneity. We used a funnel plot and Egger’s test to estimate the publication bias [49]. Additionally, we employ the trim-and-fill method for the assessment of publication bias [50]. A *p*-value less than 0.05 for Egger’s test and the asymmetry of the funnel plot are indicative of publication bias. A statistician (P.G.) on our team approved the results of the meta-analysis. We used the OpenMeta[Analyst] to perform our meta-analysis [51].

## 3. Results

### 3.1. Identification and Selection of Studies

First, we identified a total of 4489 records. Then, we removed duplicates, and 2246 records remained to review. Then, we removed 2227 records during title/abstract screening. Thus, 19 full-text articles remained eligible. Among them, we excluded 10 articles by applying our inclusion and exclusion criteria. Therefore, we included nine studies in our review [28,32,52,53,54,55,56,57,58]. We present the flowchart of the literature search in Figure 1.

### 3.2. Characteristics of the Studies

In total, 3730 nurses were included in the nine studies included in our review. The sample size ranged from 114 to 698 nurses. Most of the studies were conducted in the Republic of Korea (n = 4) and China (n = 3), while one study was conducted in Italy, and one study in Israel. The number of females was more than that of males in eight studies, while one study did not show the number of males and females. All studies used convenience samples, while eight studies were cross-sectional and one study was follow-up. Most of the nurses in the studies were recruited mainly from tertiary hospitals (n = 5), while in two studies they were mental health nurses, in one study the nurses had been working in university hospitals, and in one study the nurses had been working in a public health care unit. All studies used valid tools to measure workplace bullying, job stress, and professional quality of life. Four studies used the Negative Acts Questionnaire-Revised [59] to measure workplace bullying, while the other studies used Likert or continuous scales. Four studies measured job stress, and six studies measured professional quality of life. These six studies measured professional quality of life with the Professional Quality of Life Scale (ProQOL) [29]. The ProQOL includes three dimensions: compassion satisfaction; burnout; and secondary trauma stress. Higher scores on compassion satisfaction indicate a better professional quality of life, while higher scores on burnout and secondary trauma stress indicate a worse professional quality of life. Four studies provided response rates. We present the main characteristics of the nine studies included in our review in Table 1.

### 3.3. Quality Assessment

The quality of all the studies in our review was good. Three studies [32,56,58] did not eliminate confounders, but the scholars in these studies examined mediation models between workplace bullying, job stress, and professional quality of life. Thus, their aim was not to establish a relationship between these variables but to investigate the mediating role of some variables; so, the authors did not take into account potential confounders. We present the quality of nine studies included in our review in Appendix A.

### 3.4. Meta-Analysis

Detailed results of studies included in this systematic review are shown in Table 2.

#### 3.4.1. Workplace Bullying and Job Stress

Three studies [52,54,58] reported a correlation coefficient between workplace bullying and job stress. The correlation coefficients ranged from 0.31 to 0.37 and were statistically significant (*p* < 0.001 in all cases). The pooled correlation coefficient was 0.34 (95% CI: 0.29 to 0.39) (Figure 2); thus, a moderate positive correlation between workplace bullying and job stress was identified. There was no heterogeneity between results (I^2^ = 0%, *p*-value for the Hedges Q statistic = 0.59). A leave-one-out sensitivity analysis showed that our results were stable when we excluded one study each time. The pooled correlation coefficient varied between 0.32 (95% CI: 0.25 to 0.38, I^2^ = 0%) and 0.36 (95% CI: 0.29 to 0.43, I^2^ = 0%). The Egger’s test (*p* = 0.95) and funnel plot (Appendix A) results suggested the absence of publication bias. In the same way, the missing studies imputed by the trim-and-fill method were zero. The meta-regression analysis showed that data collection time (*p* = 0.304), females’ percentage (*p* = 0.371), age (*p* = 0.413), sample size (*p* = 0.996), and response rate (*p* = 0.924) did not affect the pooled coefficient between workplace bullying and job stress.

Only one study [52] performed a multivariable linear regression analysis and found a positive relationship between workplace bullying and job stress (unstandardized coefficient beta = 0.06, *p* < 0.001).

Moreover, one study [55] considered job stress as a dichotomous variable (high vs. low levels of job stress) and performed a multivariable logistic regression analysis. The authors considered physical aggression and non-physical aggression as independent dichotomous variables, and they found a statistically significant relationship between non-physical aggression and job stress (odds ratio = 1.81, *p* < 0.01), and a non-statistically significant relationship between physical aggression and job stress (odds ratio = 1.18, *p* > 0.01).

#### 3.4.2. Workplace Bullying and Professional Quality of Life

Four studies [32,54,56,57] measured correlation coefficients between workplace bullying and professional quality of life by using the ProQOL. Since the ProQOL measures three dimensions of professional quality of life (i.e., compassion satisfaction, job burnout, and secondary traumatic stress), we performed three meta-analyses. Kim et al. (2019) [57] did not present data for secondary traumatic stress.

Regarding compassion satisfaction, the correlation coefficients ranged from −0.13 to −0.43 and were statistically significant in three out of four studies. The pooled correlation coefficient was −0.28 (95% CI: −0.41 to −0.15) (Figure 3); thus, a small negative correlation between workplace bullying and compassion satisfaction was identified. In that case, the heterogeneity between the results was high (I^2^ = 83.0%, *p*-value for the Hedges Q statistic < 0.001). A leave-one-out sensitivity analysis showed stability for the results since the pooled correlation coefficient varied between −0.23 (95% CI: −0.36 to −0.09, I^2^ = 77.0%) and −0.32 (95% CI: −0.45 to −0.18, I^2^ = 77.2%). The Egger’s test (*p* = 0.47) and funnel plot (Appendix A) results suggested the absence of publication bias. Similarly, the missing studies imputed by the trim-and-fill method were zero. The meta-regression analysis identified that more recent studies showed a moderate negative correlation between workplace bullying and compassion satisfaction, while earlier studies showed a small negative correlation (coefficient beta = −0.06, 95% CI = −0.12 to −0.01, *p* < 0.001). Moreover, females’ percentage (*p* = 0.156) and sample size (*p* = 0.122) did not affect the pooled coefficient between workplace bullying and compassion satisfaction.

Regarding job burnout, the correlation coefficients ranged from 0.16 to 0.52 and were statistically significant in three out of four studies. The pooled correlation coefficient was 0.43 (95% CI: 0.32 to 0.53) (Figure 4); thus, a moderate positive correlation between workplace bullying and job burnout was identified. We found that the heterogeneity between the results was high (I^2^ = 80.0%, *p*-value for the Hedges Q statistic < 0.001). Our results were stable, since in the leave-one-out sensitivity analysis the pooled correlation coefficient varied between 0.39 (95% CI: 0.24 to 0.53, I^2^ = 83.6%) and 0.49 (95% CI: 0.44 to 0.53, I^2^ = 0%). The Egger’s test (*p* = 0.22) and funnel plot (Appendix A) results suggested the absence of publication bias. Moreover, the missing studies imputed by the trim-and-fill method were zero. The meta-regression analysis showed that the association between workplace bullying and job burnout was stronger in studies with a higher percentage of females (coefficient beta = 0.01, 95% CI = 0.006 to 0.019, *p* = 0.001). Additionally, the data collection time (*p* = 0.421) and sample size (*p* = 0.090) did not affect the pooled coefficient between workplace bullying and compassion satisfaction.

Regarding secondary traumatic stress, the correlation coefficients ranged from 0.02 to 0.53 and were statistically significant in two out of three studies. The pooled correlation coefficient was 0.36 (95% CI: 0.11 to 0.57) (Figure 5); thus, a moderate positive correlation between workplace bullying and secondary traumatic stress was identified. We found that the heterogeneity between the results was high (I^2^ = 93.0%, *p*-value for the Hedges Q statistic < 0.001). A leave-one-out sensitivity analysis proved the stability of our results since the pooled correlation coefficient varied between 0.26 (95% CI: −0.19 to 0.61, I^2^ = 94.3%) and 0.50 (95% CI: 0.42 to 0.57, I^2^ = 51.1%). The Egger’s test (*p* = 0.07) and funnel plot (Appendix A) results suggested the absence of publication bias. Additionally, the missing studies imputed by the trim-and-fill method were zero. The meta-regression analysis identified that more recent studies showed a moderate positive correlation between workplace bullying and secondary traumatic stress, while earlier studies showed a small positive correlation (coefficient beta = 0.08, 95% CI = −0.04 to 0.20, *p* = 0.003). Additionally, we found that the association between workplace bullying and secondary traumatic stress was stronger in studies with a higher percentage of females (coefficient beta = 0.01, 95% CI = 0.007 to 0.020, *p* = 0.002). The sample size (*p* = 0.082) did not affect the pooled coefficient between workplace bullying and secondary traumatic stress.

Four studies [28,53,54,57] conducted multivariable linear regression analyses to examine the relationship between workplace bullying and professional quality of life. Multivariable analyses confirmed the results from the correlation coefficients above. In particular, two studies [53,57] found a positive relationship between workplace bullying and job burnout (unstandardized coefficient betas were 0.03 [*p* = 0.037], and 0.15 [*p* < 0.01]). Moreover, one study [53] found a positive relationship between workplace bullying and secondary traumatic stress (unstandardized coefficient beta = 0.09 [*p* < 0.031].

## 4. Discussion

To the best of our knowledge, this is the first systematic review and meta-analysis on the association between workplace bullying caused by colleagues, job stress, and the professional quality of life of nurses. This review included nine studies involving 3730 nurses. We focused our review on workplace bullying caused by colleagues rather than on workplace bullying caused by patients and/or relatives. Our choice is guided by the fact that several systematic reviews, scoping reviews, and umbrella reviews have already investigated the impact of workplace bullying caused by patients and/or relatives on nurses [40,41,42,43,44]. Thus, we summarized the evidence on the association between workplace bullying caused by colleagues, job stress, and professional quality of life among nurses.

Our findings suggest that higher levels of workplace bullying are associated with higher levels of job stress and a worse professional quality of life for nurses. In particular, we found a moderate positive correlation between workplace bullying and job stress, job burnout, and secondary traumatic stress in nurses. Moreover, we found a small negative correlation between workplace bullying and compassion satisfaction. A meta-regression analysis showed that the association between workplace bullying and compassion satisfaction was stronger in more recent studies than in earlier studies. A similar result was found regarding the association between workplace bullying and secondary traumatic stress. Additionally, we found that the positive correlation between workplace bullying, job burnout, and secondary traumatic stress was stronger in studies with a higher percentage of females.

Workplace bullying is a frequent, dynamic, and multivariate social phenomenon, which is characterized by complexity and diversity in definitions, types, and consequences. Working in understaffed nursing departments with increased demands of care and overtime work, combined with a lack of support from leadership as well as from colleagues, influences the likelihood of developing occupational stress in nurses [60,61,62,63]. The above work-related factors, including occupational stress, form the underlying basis for the occurrence of bullying among nurses [64]. Organizational factors are seen to feed a vicious cycle of burdening nurses, where work stress plays a direct and indirect role in causing incidents of bullying. Nurses are usually threatened by burnout, low professional quality of life, and occupational stress, with numerous negative effects, not only physical and mental individual disorders but also organizational ones, such as decreased productivity, poor patient outcomes, and increased healthcare-related costs [13,16,24]. Among the various healthcare professions, it has been seen that nursing is the most vulnerable profession of all [8].

Nurses, as frontline healthcare professionals, play a key role in providing quality and safe care to patients, while ensuring their satisfaction with the care provided [65,66]. Additionally, although the quality and outcomes of care depend on the cooperation between doctors and nurses, their relationship is often disturbed by bullying [67]. In particular, the issue of patient safety has emerged as fundamental, as a significant proportion of hospitalized patients experience an adverse event with consequences for their status of health, or even their lives [67]. Nurses’ job stress, job burnout, and secondary traumatic stress, which were found in this study to be correlated with increased workplace bullying, negatively affect nurses’ performance and the quality and safety of their care, as well as patient satisfaction [68,69,70]. The above three situations experienced by nurses have a negative impact on them, affecting their physical and mental well-being as well as their performance. Musculoskeletal problems, severe sleep disturbances, depression, and anxiety, reduced work capacity, social support, and control of work, as well as increased emotional stress, quiet quitting, and work time, make up the effects of the above factors on nurses [71,72,73]. In many cases, stress, burnout, and post-traumatic stress disorder are correlated among themselves, and the presence and increased level of one feeds into the others, creating a highly stressful and particularly unhealthy work environment [74,75,76,77]. Bullying has a direct effect on both nurses and patients. Nurses who are bullied have poorer physical and mental health and experience more stress and less resilience, compared to nurses who are not bullied [78]. Bullying affects working relationships by reducing teamwork and communication and increases the likelihood of mistakes occurring during the provision of nursing care [34]. A common phenomenon is the bullying of nursing students or newly appointed nurses by more experienced nurses, which leads to new nurses feeling emotional distress, anxiety, or depression, which in turn impacts job satisfaction, cynicism, burnout, and intention to leave [79,80].

The findings of our meta-analysis showed the existence of a negative correlation between workplace bullying and compassion satisfaction. Compassion satisfaction is a positive aspect of professional quality of life. People working in the helping professions, such as nurses, derive satisfaction from the contribution they make to patients, the positive feelings they have for their colleagues, and a positive feeling resulting from the ability to help others and make a contribution. Compassion satisfaction is playing a protective role in nurses’ mental health. Specifically, when nurses experience a high level of compassion satisfaction, the likelihood of developing compassion fatigue, burnout, stress, and depression is reduced [81,82]. Additionally, through compassion satisfaction, nurses improve their performance, increase their commitment to the organization, and enhance their competencies [83]. Strengthening cooperation and good working relationships enhances compassion satisfaction and could also reduce the likelihood of individuals developing bullying behaviors [84,85].

An interesting finding of the present study is that the association of workplace bullying with compassion satisfaction was stronger in more recent studies. As the work environment of nurses is associated with the occurrence of bullying, this finding can be interpreted through the shaping of nurses’ work environments in recent years. Regarding the occurrence of bullying, the main organizational and workplace factors that predict its occurrence include inadequate staffing, lack of time to get the job done, lack of breaks away from the work, workload, and autocratic, unsupportive, and disengaged leadership [86,87]. Recently, the impact of the pandemic on health systems has created particularly challenging working conditions, favorable to the evolution of the phenomenon of bullying [88,89], as well as reduced job and compassion satisfaction [90,91,92].

The significantly high incidence of bullying, and its consequences, make it imperative that preventive measures and management actions are taken. A first approach to the prevention of the occurrence of the phenomenon by the administrations of healthcare organizations is to move towards improving the working environment of nurses, by ensuring access to the necessary resources, both material and human, and improving organizational support for nurses in their demanding working environment. Despite the increased incidence of bullying, nurse supervisors often fail to implement effective ways of managing it, instead choosing confrontation, leaving the organization, or avoiding the bullying incident [93]. In contrast, nurses suggest that effective strategies include zero tolerance for bullying, promoting teamwork and communication, establishing a supportive culture, promptly investigating the incident and confronting the bully, and creating policies around workplace behavior [94]. The implementation of training programs, cognitive rehearsal programs, and educational programs can reduce the impact of bullying on nurses in the workplace [95]. The need to educate nurses regarding the management of bullying is highlighted by the fact that often nurses choose negative coping strategies, such as avoidance, resorting to substance use, and exhibiting lower levels of acceptance, as their primary coping mechanisms [96]. When nurses use positive coping strategies and resilience, the impact of bullying on the quality of their work life is mediated [32,56].

### Limitations

Although we performed a rigorous systematic review and meta-analysis following PRISMA guidelines, our study had some limitations. Firstly, the number of studies included in our meta-analysis was small. Thus, we could not further explore potential sources of heterogeneity, such as country, study design, sampling method, clinical settings, assessment tools, and quality of studies. However, we performed a meta-regression analysis for several variables, namely, data collection time, females’ percentage, age, sample size, and response rate. Secondly, seven out of the nine studies in this review were conducted in the Republic of Korea (n = 4) and China (n = 3). Additionally, the majority of our studies included nurses in hospitals. Thus, the generalization of our findings should be made with caution. Further studies in different countries, clinical settings, and working conditions should be conducted to expand our knowledge. Thirdly, there were limited data from studies that explored the independent impact of workplace bullying on job stress and professional quality of life by constructing multivariable models and eliminating confounders. Thus, we cannot perform a meta-analysis using adjusted coefficients beta to improve the validity of our findings. However, we performed four meta-analyses by using the correlation coefficients. Moreover, our leave-one-out sensitivity analyses suggested the stability of our results. Future studies should emphasize the use of multivariable models to eliminate confounding. Fourthly, eight out of nine studies in this review were cross-sectional studies. Therefore, we could not establish a causal relationship between workplace bullying, job stress, and the professional quality of life of nurses. Follow-up studies should be conducted to further explain the association between our study variables. Fifthly, all the included studies used convenience samples to recruit nurses. This approach introduced selection bias. For example, the percentage of females was higher in all studies. Future studies should employ random and stratified samples to produce more valid results. Finally, we employed five major databases to perform our review. Moreover, we searched for articles in the English language. Therefore, it is probable that we missed studies in this review. However, our meta-analyses suggested the absence of publication bias.

## 5. Conclusions

Our meta-analysis suggests that workplace bullying has a moderate positive correlation with job stress, job burnout, and secondary traumatic stress in nurses. Additionally, there is a small negative correlation between workplace bullying and compassion satisfaction. Thus, higher levels of workplace bullying are associated with higher levels of job stress and worse professional quality of life. Our study emphasizes the necessity for immediate solutions through guidelines for better identification, prophylaxis, and management of workplace bullying, as well as well-organized educational programs. Our findings may prompt nursing managers and policy-makers to care about how nurses interact and work with their colleagues. Ensuring a healthy working environment, where nurses have access to the necessary resources and organizational support, eliminates the development of bullying. Policy-makers must take into consideration that there is an urgent need for safety and prevention strategies to be established as well as suitable educational programs to be promoted.

## Figures and Tables

**Figure 1 healthcare-12-00623-f001:**
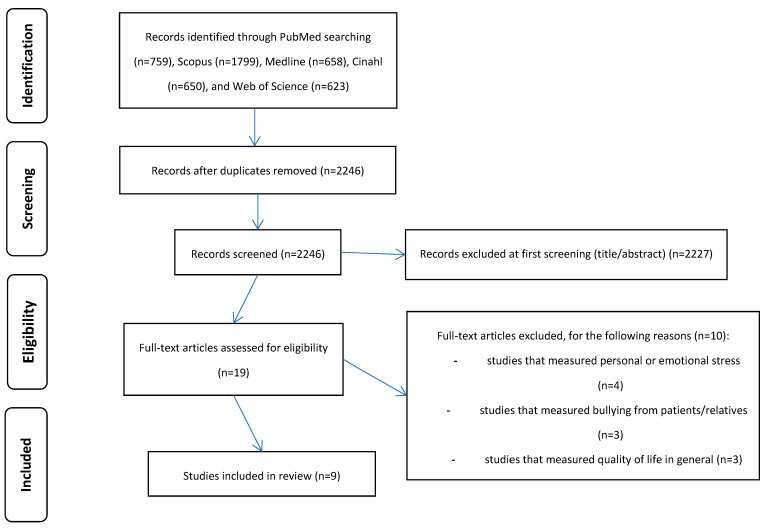
Flowchart of the systematic review.

**Figure 2 healthcare-12-00623-f002:**
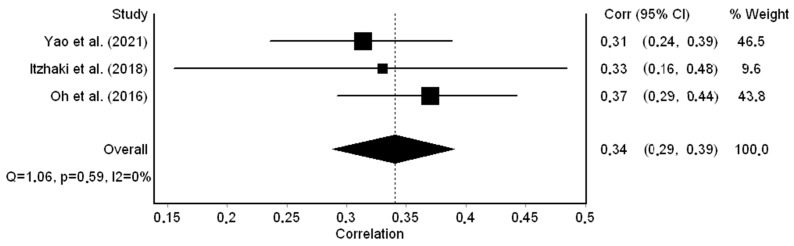
Forest plot for the correlation coefficient between workplace bullying and job stress [52,54,58].

**Figure 3 healthcare-12-00623-f003:**
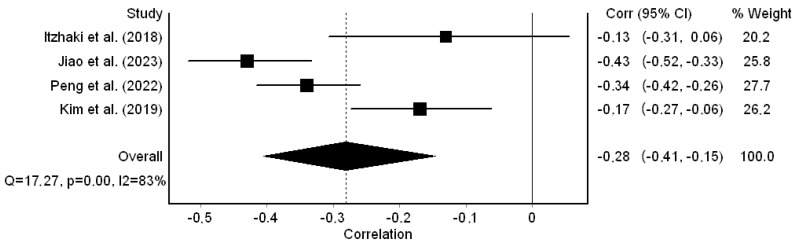
Forest plot for the correlation coefficient between workplace bullying and compassion satisfaction [32,54,56,57].

**Figure 4 healthcare-12-00623-f004:**
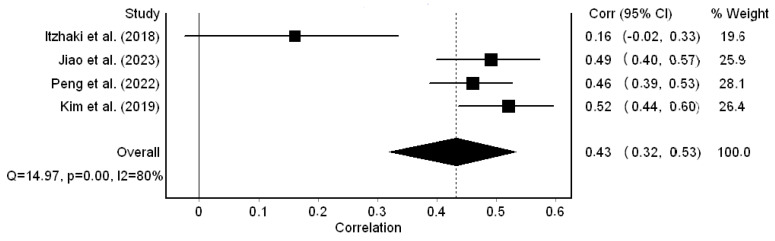
Forest plot for the correlation coefficient between workplace bullying and job burnout [32,54,56,57].

**Figure 5 healthcare-12-00623-f005:**
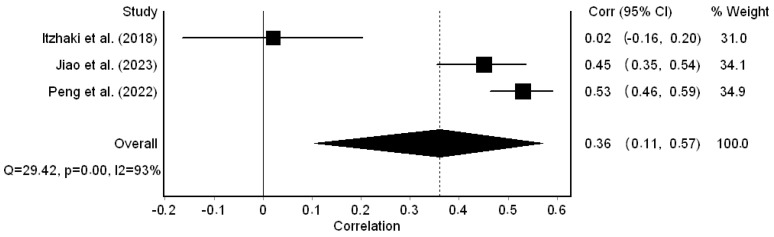
Forest plot for the correlation coefficient between workplace bullying and secondary traumatic stress [32,54,56].

**Table 1 healthcare-12-00623-t001:** Main characteristics of studies included in this systematic review.

Reference	Country	Data Collection Time	Females (%)	Age, Mean (SD)	Sample Size (n)	Study Design	Sampling Method	Clinical Settings	Assessment Tool for Workplace Bullying	Assessment Tool for Job Stress	Assessment Tool for Professional QoL	Response Rate (%)
Yao et al. [52]	China	2019	73.5	35.9 (8.0)	539	Cross-sectional	Convenience	Mental health services	Workplace violence scale	Chinese nursing stress scale	None	97.8
Kwak et al. [53]	Republic of Korea	2016	NR	<30 years, 66.4%	399	Cross-sectional	Convenience	University hospitals	Workplace violence scale	None	ProQOL	NR
Itzhaki et al. [54]	Israel	2018	56.3	47.3 (9)	114	Cross-sectional	Convenience	Mental health services	5-point Likert scale	5-point Likert scale	ProQOL	NR
Magnavita [55]	Italy	2003–2009	64.0	38.9 (8.7)	698	Follow-up	Convenience	Public health care unit	Violent incidents	Continuous scale	None	96.5
Jiao et al. [56]	China	2022	92.6	≤35 years, 88.6%	297	Cross-sectional	Convenience	Tertiary hospitals	NAQR	None	ProQOL	97.1
Peng et al. [32]	China	2021	93.5	≤35 years, 44.4%	493	Cross-sectional	Convenience	Tertiary hospitals	NAQR	None	ProQOL	NR
Kim et al. [57]	Republic of Korea	2018	96.3	NR	324	Cross-sectional	Convenience	Tertiary hospitals	NAQR	None	ProQOL	NR
Choi & Lee [28]	Republic of Korea	2015	96.9	NR	358	Cross-sectional	Convenience	Tertiary hospitals	Verbal and physical violence, and physical threats	None	ProQOL	NR
Oh et al. [58]	Republic of Korea	2013	97.2	25.6 (3.8)	508	Cross-sectional	Convenience	Tertiary hospitals	NAQR	Occupational stress scale-short form	None	95.8

NAQR: Negative Acts Questionnaire-Revised; NR: not reported; ProQOL: professional quality of life; QoL: quality of life.

**Table 2 healthcare-12-00623-t002:** Detailed results in studies included in this systematic review.

Reference	Association between Workplace Bullying and Job Stress			Association between Workplace Bullying and Professional QoL	
	Correlation Coefficient (*p*-Value)	Regression Coefficient Beta (*p*-Value)	Odds Ratio (*p*-Value)	Correlation Coefficient (*p*-Value)	Regression Coefficient Beta (*p*-Value)
Yao et al. [52]	0.31 (<0.001)	0.06 (<0.001)			
Kwak et al. [53]					CS: 0.03 (0.07); B: 0.03 (0.037); STS: 0.09 < 0.001
Itzhaki et al. [54]	0.33 (<0.001)			CS: −0.13 (>0.05); B: 0.16 (>0.05); STS: 0.02 (>0.05)	CS: NR (>0.05); B: NR (>0.05); STS: NR (>0.05)
Magnavita [55]			Physical aggression: 1.18 (>0.05); Non-physical aggression: 1.81 (<0.01)		
Jiao et al. [56]				CS: −0.43 (<0.001); B: 0.49 (<0.001); STS: 0.45 (<0.001)	
Peng et al. [32]				CS: −0.34 (<0.01); B: 0.46 (<0.01); STS: 0.53 (<0.01)	
Kim et al. [57]				CS: −0.17 (<0.01); B: 0.52 (<0.01)	CS: 0.02 (0.65); B: 0.15 (<0.01)
Choi & Lee [28]					CS: 0.05 (>0.05); B: 0.03 (>0.05); STS: 0.06 (>0.05)
Oh et al. [58]	0.37 (<0.001)				

B: Burnout; CS: compassion satisfaction; NR: not reported; QoL: quality of life; STS: Secondary traumatic stress.

## Data Availability

Data will be available after a reasonable request from the authors.

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
