# Peer review of "Association between Workplace Bullying, Job Stress, and Professional Quality of Life in Nurses: A Systematic Review and Meta-Analysis"

_healthcare, 2024, doi:10.3390/healthcare12060623_

Round 1
Reviewer 1 Report
Comments and Suggestions for Authors
This is a critical study, and the authors are congratulated for conducting this work. I recommend several points that need to be considered in your manuscript. I hope it might help you with the improvement.
The abstract does not address the background and rationale of the study but only the objectives and the gap. Please integrate.
The introduction section is very well done but strangely does not address in a general way the issues related to the risk factors of job stress in nurses and then arrive deductively at workplace bullying (please see: DOI: 10.1590/1516-3180.2021.0068.R1.31052021; and DOI: 10.1097/NMD.0000000000001504).
Some references must be added to the introduction section (e.g., line 41).
It needs to be specified whether a statistician has approved the results.
In the discussion section, some references should be improved (e.g., lines 297 and 308). Only some evaluation parameters resulting from the described bullying consequences are addressed (No proposal of a different scale or criteria to quantify this issue).
No proposals are made for the future to contrast this phenomenon.
The conclusions could be more specific; for example, what did the authors intend with the phrase in lines 398-399?
What is intended with "good working"? How could the hospital facilities improve the nurses' safety and mental health?
Please revise the figure 1 and its coherence with the text.
Author Response
Dear Reviewer,
Please, see the attached file.

Reviewer 2 Report
Comments and Suggestions for Authors
Dear Researchers,
I enjoyed reading your manuscript on “Association between workplace bullying, and job stress and professional quality of life in nurses: a systematic review and meta-analysis” which can provide an understanding of the workplace bullying and violence among nurses. I suggest some revisions on your paper to make it better.
1. 1. Acute stress, the most common, happens when somebody cannot easily handle situations that an average person can, episodic stress occurs when acute stress takes place frequently and chronic stress occurs when the period of the phenomenon occurrence is long lasting. Line 61, 62, 63, occurs/happens missing
2. Please provide a little bit more details as to why the research looked at study on workplace bullying caused by colleagues rather than on workplace bullying caused by patients and/or relatives. Line 85-87. Was there previous systematic reviews already published on workplace bullying caused by patients and/or relatives? Then details could include some background in the introduction section on previous systematic reviews published in workplace bullying caused by patients and relatives and data findings from those studies. Please include in discussions how workplace bullying data in your article caused by colleagues is different from any previous studies on workplace bullying caused by patients and relatives.
3. In the Materials and Methods section, please provide from when to Jan 04, 2024 articles were searched (for 10 years or 20 years ….)? We searched PubMed, Medline, Scopus, Cinahl, and Web of Science up to January 04, 2024. Line 91
4. Since the articles chosen for the review were from different parts of the world including Republic of Korea, China, Italy, and Israel. I suggest giving an annual cost estimates from some of those countries mentioned. Only United States is mentioned although the articles reviewed were not chosen from US. It was estimated that in the United States, the 58 annual cost of job stress is about $ 200 – 300 million Line 58.
5. Flowchart of the systematic review Figure 1- some of the fonts are not readable, some fonts are cutoff/missing. Font size should be made big enough and clear for readers to read.
6. Please include in the discussion section what other previous systematic studies on job stress and workplace bullying in nursing have shown and how your study is similar or differ to the other studies conducted. Explain how your findings fit with existing literature. Describe how they extend the findings of previous studies. Discuss whether your results support or contradict findings shown in other systematic review studies. If you find contradictions, explain why you think your results are different from what has already been shown. Did you answer a similar question with a different model? Did you assess a different timeline or use a different strategy?
Comments on the Quality of English LanguageEnglish language is fine. Minor editing required.
Author Response

(The authors gave the same response as above.)

Reviewer 3 Report
Comments and Suggestions for Authors
The evaluated manuscript is a meta-analysis focused on the association between workplace bullying, and job stress and professional quality of life in nurses. The topic is highly relevant, and the findings can serve healthcare service and hospitals managers in the design and implementation of preventive policies against workplace bullying towards nurses.
Overall, the introduction is well-written, well-structured, and provides a concise summary of the subject matter. Both the methodology and results sections are rigorous and exhaustive, although I have some suggestions:
In the introduction, there should be an argumentation on why the potential moderating variables were chosen. Furthermore, there should be a precise definition of what exactly constitutes compassion satisfaction.
The research questions should be formulated.
Newer PRISMA guidelines exist.
The number of individuals performing the data extraction procedure and the resolution process for any discrepancies should be reported.
Reformat the flowchart as many of the steps are currently illegible.
The absence of funnel plots in the supplementary material is noted. Additionally, and given the limited number of included studies, it would be advisable to consider employing an alternative method, in addition to funnel plots and the Egger's test, to examine the risk of publication bias.
Including the PRISMA checklist in the supplementary material would be beneficial.
Author Response

(The authors gave the same response as above.)

Round 2
Reviewer 2 Report
Comments and Suggestions for Authors
Dear Authors,
Please review some minor comments below. Thank you.
1. Line 929 Among the various healthcare professions, it has been seen that nurses is the most vulnerable profession of all. Should be nursing.
2. Line- 1224 Ensuring a healthy working environment, where nurses have access to the necessary resources and organizational support reduces the chances of bullying. developing. Eliminate developing.
3. Line 1133-1134 Despite the increased incidence of bullying, nurse supervisors often fail to implement effective ways of managing it, choosing confrontation, leaving the organization and avoidance avoiding the problem. [96]. Just keeping everything similar in the sentence such as choosing, leaving, avoiding.
4. In the discussion section, lines 1128-1140, it will be helpful to include information about creating counselling programs to help nurses cope with stress related to workplace violence (as discussed in your chosen articles).
5. When your study has focused on workplace violence perceived by colleagues, it will be helpful to include some information in the discussion section about physician-nurses relationship and experienced vs young nurses relationship. (one of the study in your manuscript mentioned some details https://doi.org/10.1111/ijn.12792)
Comments on the Quality of English LanguageQuality of English is fine. Minor editing required.
Author Response
Dear Reviewer 2,
Thank you very much for the peer review of the paper “Association between workplace bullying, and job stress and professional quality of life in nurses: a systematic review and meta-analysis” and your comments, which have improved the quality of the manuscript.
Please, see the attached file
